# Prevalence, Risk Factors of Preserved Ratio Impaired Spirometry in adult in plateau: A Cross-Sectional Study

**Junjie Xia**[1][☮]*, **Yu Qiu**[1][☮], **Ling Huang**[1], **Wenjun Li**[1], **Xingxiong Zou**[2], **Xiaoqian Wang**[1], **Xiaoli Hou**[1], **Yuting Chen**[1], **Haishi Xue**[3], **Runjiao Chen**[3], **Lingna Li**[3], **Ke Yi**[4], **Jincheng Wang**[5], **Wenying Xie**[1]

1 Department of Respiratory and Critical Care Medicine, The Third Hospital of Mianyang/Sichuan Mental Health Center, Mianyang, Sichuan, China, 2 Department of Radiology, The Third Hospital of Mianyang/Sichuan Mental Health Center, Mianyang, Sichuan, China, 3 Clinical Medical Sciences of Southwest Medical University, Luzhou, Sichuan, China, 4 Department of Respiratory and Critical Care Medicine, Sichuan Science City Hospital, Mianyang, Sichuan, China, 5 Department of Respiratory and Critical Care Medicine, The people's hospital of zhongjiang, Deyang, Sichuan, China

☮ These authors contributed equally to this work.
* 382044970@qq.com

## Abstract

### Background

Preserved ratio impaired spirometry(PRISm) is considered to be a precursor of COPD. The purpose of our study is to investigate the prevalence and risk factors of PRISm in high-altitude areas.

### Methods

The adult residents of Hongyuan County were selected by random sampling method, and the lung function tests, questionnaires, blood tests were conducted. The prevalence of PRISm was compared among different factors of investigation, and binary logistic regression analysis was used to determine the independent influencing factors of PRISm.

### Results

627 participants qualified for quality control, the prevalence was 10.06%. The independent factors of PRISm were age 40–49 years old(OR = 4.322,95%CI: 1.149–16.262),age≧60 years (OR = 4.453, 95% CI: 1.003–19.762),Body mass index≧30(OR: 3.745, 95% CI: 1.611–8.707),Smoking (OR: 2.591, 95% CI: 1.305–5.146), Diabetes (OR: 3.894, 95% CI: 1.043–14.199), history of pulmonary tuberculosis (OR: 13.678, 95% CI: 5.495–34.049), hypertension(OR: 3.447, 95% CI: 1.529–7.771), White blood cell count(OR: 1.414, 95% CI: 1.164–1.717), and Red blood cell volume distribution width (OR: 1.098, 95% CI: 1.009–1.196).

### Conclusion

The prevalence of PRISm in Hongyuan County was 10.03%; The independent influencing factors of PRISm included age, smoking, the history of tuberculosis, diabetes,hypertension,body mass index≧30,Red blood cell volume distribution width.

**Data availability statement:** Participant confidentiality restrictions prohibit the authors from making the data set publicly available. The Third Hospital of Mianyang's Ethics Committee approved this study. Any queries about the data may be directed to the Third Hospital of Mianyang's Research Governance by contacting the secretary, Xin Shen Li (578012475@qq.com).

**Funding:** This study was supported by the Medical Research project of Sichuan Medical Association [S2024061 to JX], the Scientific Research Project of Mianyang Municipal Health Commission [202343 to JX], and the Health Commission of Sichuan Province [20PJ267 to JX]. The funders had no role in study design, data collection and analysis, decision to publish, or preparation of the manuscript.

**Competing interests:** The authors have declared that no competing interests exist.

**Abbreviations:** COPD, Chronic Obstructive Pulmonary Disease; PRISm, Preserved ratio impaired spirometry; FEV1, Forced expiratory volume in 1s; FVC, Forced vital capacity; WBC, White blood cell count; RDW, Red blood cell volume distribution width; BMI, Body mass index.

## Introduction

Chronic Obstructive Pulmonary Disease(COPD) is considered a heterogeneous lung condition characterized by persistent airflow limitation due to pathological changes in the large and small airways. Studies [1,2] found that the prevalence of COPD was higher in high-altitude areas. A meta-analysis [3] also found that the prevalence of COPD was higher in high-altitude areas, especially in Asia. In the plateau of China, located at an altitude above 3000 meters, revealed[4] the prevalence of COPD was 12.16%, higher than the national level(9.9%).

Preserved ratio impaired spirometry(PRISm) is defined as a normal or preserved forced expiratory volume in 1s(FEV1)/forced vital capacity(FVC) ratio(≧0.7) but a FEV1 of less than 80% predicted [5]. In US adults, previous studies have shown that individuals with PRISm have an increased risk of respiratory symptoms and mortality, and about 50% of those might transition to COPD during 5 years period [6–8]. PRISm is considered to be a precursor of COPD.Therefore, understanding the prevalence and risk factors of PRISm in highland areas is helpful to prevent COPD.

In this study, a cross-sectional survey was conducted to investigate the prevalence rate and risk factors of PRISm among the adults in Hongyuan County (average altitude 3507 meters), providing evidence for the prevention of COPD in plateau areas.

## Method

### Study design

We conducted a cross-sectional study of adult population in Hongyuan County, Aba Prefecture, Sichuan Province, from June 2020 to December 2020. This study was approved by the Ethics Review Committee of Hospital.All of the survey respondents signed the informed consent form. We selected participants using a simple random sampling method from each of the gender/age strata from communities or villages,The proportion of samples from each age group was based on the 2010 census. Since there was no available research data for reference in China, we used a single proportional formula to calculate the sample size. $N = Z^2 * P (1-P)/d^2$. Then the sample size was 544 based on the total population. Considering that the loss of follow-up rate was 10%, the final sample size was 560($Z = 1.96, p = 0.15, d = 0.2p$).

### Participants

Ethics was approved by The Third Hospital of Mianyang's Ethics Committee. The study population was fulfilled the following criteria:

Inclusion criteria:living in their current residence for more than 5 years;aged ≥ 18years;lung function quality B and above; complete clinical data.

Exclusion criteria: living in functional areas such as student dormitories, army, sheds, nursing homes, temple; mental illness and cognitive impairment (dementia, comprehension disorders, deaf); high paraplegia; pregnant or lactating women; newly detected and treated tumors;neuromuscular disorders;COPD; history of lung and abdominal surgery.

### Outcome measures

**1. Pulmonary function test.** According the Guidelines for Pulmonary Function Examination, [9] we used the survey used uniform methods, procedures and US spiro-PD portable lung function instrument. Spirometry testing was undertaken by a skilled, full-time spirometry technician. Repeated 3–8 times of lung ventilation function test, with each interval > 1 min. During the measurement, the maximum difference between the FEV1 and FVC was within 0.2 L, and the best value was taken. For those with airflow limitation, mesured

again with 15–30minutes after inhaling 400ug of albuterol, and the best detection value was taken. COPD was defined as FEV1/FVC < 70% after inhalation, [10] PRISm was defined as FEV1/FVC ≥ 70% and FEV1 and/or FVC < 80% predicted [5].

**2. Clinical data.** We collected the clinical data including gender,age,hypertension,diabetes,coronary heart disease,Kaschin-Beck disease, tuberculosis history(refers to the respondents who were diagnosed with tuberculosis in a regular medical institution and have been cured for more than 1 year or more),somking(1 cigarette per day, continue smoking for 1 month, or total smoking 100).The uniformly trained respiratory physicians asked the respondents face to face and asked questions in the independent and quiet room according to the questionnaire items.At the same time, we reviewed the respondents' medical records. If there was any discrepancy between the medical records and the responses, the medical records were taken as the reference.

**3. Blood parameters.** Within 1 week after lung function test,Participants' blood parameters were measured using an automated blood cell analyzer (Sysmex XT-1000) by laboratory doctor with 10 years of experience.

The project leader supervised and controlled the whole process of the epidemiological survey.

## Statistical analysis

The data were computed in the SPSS 27.0 program. The Kolmogorov-Smirnov test was performed to determine whether the samples conformed to a normal distribution. Continuous variables conforming to a normal distribution were described by means and SD, nonnormal continuous variables by median and inter-quartile range, and categorical variables by frequency and percentage. Comparisons between continuous variables in the 2 groups were made via an independent sample t test or the rank sum test (Mann-Whitney U test) depending on whether they conformed to a normal distribution. Categorical variables were tested by $\chi^2$ test or Fisher test. Independent risk factors were identified by binary logistic regression analysis. P < 0.05was deemed statistically significant.

## Results

### Clinical information

A total of 1021 individuals were surveyed and evaluable data were available for 627 participants(Fig 1 is the flow chart of the study).The subjects included 340 males (54.23%) and 287 females (45.77%), with an average age of 46.44 ± 11.00 years old.Overall prevalence: 10.05%, male prevalence: 13.24%, female prevalence: 6.28%. The prevalence rate of Han nationality was 12.41%, and that of Tibetan nationality was 8.31%. Prevalence in people aged 40 years or older is 14.15%. 19.45% of the population total smoking rate. 3.19% had coronary heart disease, 4.15% had diabetes, 6.1% had a history of tuberculosis, 4% had Kaschin-Beck disease, and 10.85% had hypertensionb(Table 1).

### Univariate Analysis

18potential risk factors associated with PRISm were screened by univariate analysis (Table 1). Multiple colinearity between variables of blood parameter was tested through variance inflation factor (VIF), which was considered to have severe multiple colinearity between variables when the VIF was greater than10. We removed multiple colinear variables by stepwise backward logistic regression, and the final variables including Red blood cell volume distribution width(RDW) (P = 0.003), White blood cell count(WBC)(P = 0.003) and Eosinophil count(EOS)(P = 0.046) were used to screen independent risk factors,the VIF between three variables was less than 5.

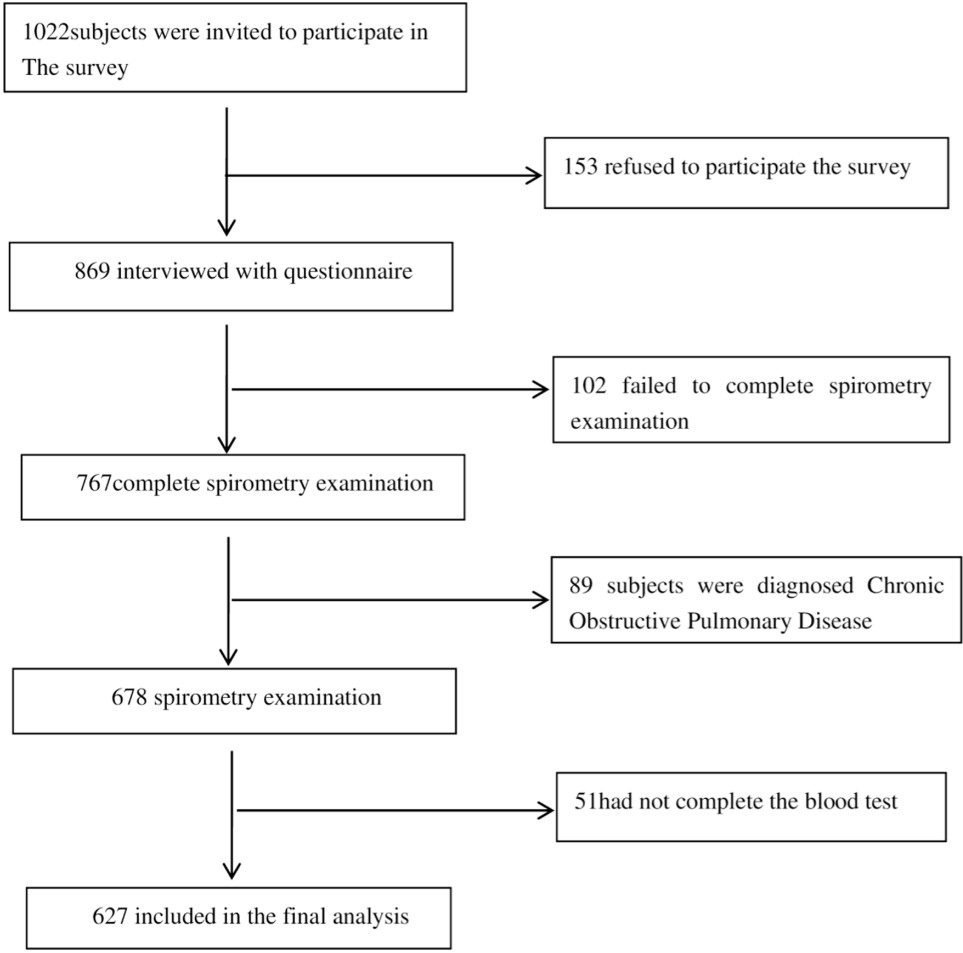

**Fig 1. Flow chart of the study.**

## Independent Risk Factors

11 variables were used to analyze the independent risk factors by Binary Logistic regression. The study showed that the independent risk factors for PRISm included age(40–49, ≧ 60),smoking,history of tuberculosis,Diabetes, Hypertension,White blood cell count of blood, RDW,BMI > 30 (Table 2).

## Discussion

The prevalence of PRISm in population-based studies ranges from7.1%to11% in Europe and the United States.[6,11,12] The incidence in Asian ranges from 8.9% to 25.2%.[13,14,15] A cross- sectional study in Malawi revealed 38.6% of participants with PRISm.[16] The different prevalence rate may be related to the different geographical and population selection. Our study first revealed that in the plateau region with an average altitude of 3,507 meters, the prevalence of PRISm among adults was 10.05%, which is higher than that of the general population(5.5%) in China.[17]

The relationship between smoking and PRISm is still controversial. Most studies have shown that smoking is a risk factor for PRISm,[11,18] however, the rate of PRISm occurrence in former smokers is not higher than that of never smokers,[11,19] even a study[20] have

**Table 1. Clinical information and univariate analysis.**

| Characteristics | Normal population (*n* = 564) | PRISm population (*n* = 63) | t/Zvalue | *P* value |
|---|---|---|---|---|
| Gender(Male/Female) | (295/269) | (45/18) | 8.35 | 0.004 |
| Smoking(Yes/No) | (101/463) | (21/42) | 8.604 | 0.003 |
| BMI(25 > BMI/30 > BMI ≥ 25/BMI ≥ 30) | (318/194/52) | (22/25/16) | 18.941 | <0.001 |
| Age(<40/40–49/50–59/ ≥ 60) | (140/232/131/61) | (3/25/21/14) | 17.952 | <0.001 |
| Nation(Han/Tibetan) | (233/331) | (33/30) | 2.843 | 0.092 |
| Coronary heart disease(Yes/No) | (14/550) | (6/57) | 6.962 | 0.008 |
| Diabetes(Yes/No) | (19/545) | (7/56) | 6.709 | 0.001 |
| History of tuberculosis(Yes/No) | (23/541) | (15/48) | 38.753 | <0.001 |
| Kaschin-Beck disease(Yes/No) | (21/543) | (4/59) | 0.450 | 0.502 |
| Hypertension(Yes/No) | (55/509) | (13/50) | 6.942 | 0.008 |
| Red blood cell count ($10^{12}$/L)[M(P25,P75)] | 5.70(5.32,6.12) | 5.75(5.49,6.32) | −1.873 | 0.061 |
| Hemoglobin(g/L)[M(P25,P75)] | 171.1(156,183.1) | 180.1(167,188.1) | −3.090 | 0.002 |
| Platelet count ($10^9$/L)[M(P25,P75)] | 213(178,257) | 202(176,250) | −0.676 | 0.499 |
| Platelet distribution width (%)[M(P25,P75)] | 0.25(0.22,0.3) | 0.25(0.21,0.28) | −1.558 | 0.119 |
| Hematokrit(%)[M(P25,P750)] | 49.8(45.93,53.09) | 51.51(48.9,55.41) | −2.939 | 0.003 |
| Mean red blood cell volume (fl)[M(P25,P75)] | 87(84.05,89.9) | 88(84.12,91.2) | −1.205 | 0.228 |
| White blood cell count ($10^9$/L)[M(P25,P75)] | 5.65(4.74,6.63) | 6.33(5.34,7.60) | −3.68 | <0.001 |
| Mean corpuscular hemoglobin(pg) [M(P25,P75)] | 29.9(28.7,30.8) | 30.3(29,31.5) | −2.208 | 0.027 |
| Neutrophil count ($10^9$/L)[M(P25,P75)] | 2.84(2.26,3.54) | 3.46(2.68,4.15) | −3.23 | 0.001 |
| Lymphocyte count ($10^9$/L)[M(P25,P75)] | 1.9(1.53,2.25) | 1.98(1.59,2.54) | −2.075 | 0.038 |
| Monocyte count($10^9$/L)[M(P25,P75)] | 0.35(0.28,0.43) | 0.36(0.34,0.46) | −2.835 | 0.005 |
| Eosinophil count ($10^9$/L)[M(P25,P75)] | 0.08(0.05,0.12) | 0.1(0.06,0.18) | −3.299 | <0.001 |
| Basophil count ($10^9$/L)[M(P25,P75)] | 0.01(0.01,0.02) | 0.02(0.01,0.03) | −2.241 | 0.025 |
| Red blood cell volume distribution width, RD(fl) [M(P2,P75)] | 43.4(41.7,45.1) | 45.1(42.4,46.9) | −3.28 | 0.001 |

BMI:Body mass index; PRISm:Preserved ratio impaired spirometry;RDW:Red blood cell volume distribution width

found that smoking was negatively correlated with PRISm occurrence, indicating that smoking is not completely related to the incidence of PRISm.[21] This study was consistent with most previous studies.

Advanced age and diabetes are considered risk factors for PRISm.[22,23] Diabetes is associated with lower FEV1 and FVC, but not with FEV1/FVC.[24] Some circulating metabolites such as glycoprotein acetyl may play a mediating role in the association.[25] Interesting,our study found that risk factors for PRISm in high-altitude areas include the age groups of 40–49 years and ≥ 60 years, possibly due to a higher smoking rate in the 40–49 age group(40%) compared to the 50–59 age group(28.5%).

Higher BMI is a risk factor for the development of PRISm,[8,12] but a study[18] found that the incidence of PRISm did not increase with the increase of body weight. A study from China[26] showed that BMI < 18.5 kg/m² and BMI ≥ 35 kg/m² were risk factors for PRISm, and BMI between 28 and 34.9 kg/m² was protective factor for PRISm.Our study found that BMI greater than 30 was a risk factor for PRISm in plateau areas, probably because patients with higher BMI are more likely to have restrictive ventilation dysfunction, and the low-oxygen environment at plateau makes this phenotype more obvious.

Hypertension was also a risk factor for PRISm.[13] In plateau areas, the prevalence of hypertension is relatively high, and hypertension is a risk factor for PRISm. People with a

**Table 2.  Binary Logistic regression.**

| Variables | β | Standard Error | waldX² | OR | 95%CI | P |
|---|---|---|---|---|---|---|
| Gender | | | | | | |
| Female | | | | 1.000 | | |
| Male | 0.648 | 0.362 | 3.198 | 1.911 | 0.279–1.432 | 0.074 |
| Age | | | | | | |
| <40 | | | | 1.000 | | |
| 40–49 | 1.464 | 0.676 | 4.687 | 4.322 | 1.149–16.262 | 0.030 |
| 50–59 | 1.301 | 0.692 | 3.53 | 3.672 | 0.945–14.266 | 0.060 |
| ≧60 | 1.494 | 0.760 | 3.859 | 4.453 | 1.003–19.762 | 0.049 |
| BMI | | | | | | |
| 25 > BMI | | | | 1.000 | | |
| 30 > BMI ≧ 25 | 0.394 | 0.364 | 1.172 | 1.483 | 0.726–3.029 | 0.279 |
| BMI ≧ 30 | 1.320 | 0.430 | 9.408 | 3.745 | 1.611–8.707 | 0.002 |
| Coronary heart disease | 0.506 | 0.672 | 0.568 | 1.659 | 0.445–6.186 | 0.451 |
| Smoking | 0.952 | 0.350 | 7.395 | 2.591 | 1.305–5.146 | 0.007 |
| Diabetes | 1.348 | 0.666 | 4.094 | 3.849 | 1.043–14.199 | 0.043 |
| History of tuberculosis | 2.616 | 0.465 | 31.603 | 13.678 | 5.495–34.049 | <0.001 |
| Hypertension | 1.237 | 0.415 | 8.899 | 3.447 | 1.529–7.771 | 0.003 |
| White blood cell count | 0.346 | 0.099 | 12.211 | 1.414 | 1.164–1.717 | <0.001 |
| RDW | 0.094 | 0.044 | 4.651 | 1.098 | 1.009–1.196 | 0.031 |
| Eosinophil count | 0.062 | 0.038 | 2.687 | 1.063 | 0.988–1.145 | 0.101 |

BMI:Body mass index; PRISm:Preserved ratio impaired spirometry;RDW:Red blood cell volume distribution width

history of tuberculosis(TB) have more than twice the risk of airflow obstruction than people without a history of TB, and this association is more pronounced in low/middle income areas.[27] This study also reached a similar result, which may be related to the high incidence of TB and low income in the plateau region.

We found that the WBC and RDW were independent risk factors for PRISm population in plateau area. Individuals in the normal to PRISm trajectory and persistent PRISm trajectory had a higher WBC than those in the normal trajectory and PRISm to normal trajectory [28]. The increase of WBC was associated with the rapid decline of $FEV_1$.[29] RDW indicates systemic hypoxic load, especially in pulmonary conditions.[30] Hypobaric hypoxia increased RDW.[31] However, RDW negatively correlated with FEV1.[32] Studies[33,34,35] have shown that RDW can reflect chronic inflammation in patients with COPD and pulmonary hypertension, and is positively correlated with C-reactive protein and nterleukin-6.The increase of WBC and RDW in PRISm may be the result of the dual effect of hypoxia and chronic inflammation.

In highland areas, we can prevent prism development by controlling weight, smoking, diabetes, and high blood pressure. The history of tuberculosis may be a unique risk factor for PRISm in the plateau region of China, we can prevent the development of PRISm by controlling the incidence of tuberculosis. PRISm can also be detected early by focusing on the WBC and RDW of the population.

There are some limitations in this study. First, the geographical scope of the study is small and the sample size is small. Secondly, due to climatic reasons, some people aged ≧70 years left their permanent residence, so the actual sample of this group is small, which may lead to an underestimate of the prevalence rate. Third, such cross-sectional surveys do not eliminate recall bias.

## Conclusion

It was concluded that the prevalence of PRISm in Hongyuan County was 10.03%; The independent influencing factors of PRISm including age(40–49 years old,age≧60 years), smoking, the history of tuberculosis,diabetes,hypertension,BMI≧30,WBC, RDW in the plateau, are similar to those in plain areas.

## Author contributions

**Conceptualization:** Junjie Xia, Yu Qiu, Ling Huang, Wenjun Li, Xingxiong Zou, Xiaoli Hou, Yuting Chen, Haishi Xue, Jincheng Wang.

**Data curation:** Junjie Xia, Yu Qiu, Ling Huang, Wenjun Li, Xingxiong Zou, Xiaoqian Wang, Xiaoli Hou, Yuting Chen, Runjiao Chen, Lingna Li, Ke Yi.

**Formal analysis:** Junjie Xia, Yu Qiu, Ling Huang, Wenjun Li, Xingxiong Zou, Xiaoqian Wang, Xiaoli Hou, Yuting Chen.

**Funding acquisition:** Junjie Xia, Yu Qiu, Xingxiong Zou.

**Investigation:** Junjie Xia, Yu Qiu, Ling Huang, Wenjun Li, Xingxiong Zou, Xiaoqian Wang, Xiaoli Hou, Haishi Xue, Runjiao Chen, Wenying Xie.

**Methodology:** Junjie Xia, Yu Qiu, Ling Huang, Wenjun Li, Xingxiong Zou, Xiaoqian Wang, Xiaoli Hou, Yuting Chen, Haishi Xue, Runjiao Chen, Wenying Xie.

**Project administration:** Junjie Xia, Yu Qiu, Ling Huang, Wenjun Li, Xingxiong Zou, Xiaoqian Wang, Xiaoli Hou, Yuting Chen, Runjiao Chen, Ke Yi, Jincheng Wang.

**Resources:** Junjie Xia, Yu Qiu, Wenjun Li, Xingxiong Zou, Xiaoqian Wang, Xiaoli Hou, Runjiao Chen, Lingna Li, Wenying Xie.

**Software:** Junjie Xia, Yu Qiu, Ling Huang, Wenjun Li, Xingxiong Zou, Xiaoqian Wang, Xiaoli Hou, Yuting Chen, Haishi Xue, Runjiao Chen, Lingna Li, Ke Yi, Jincheng Wang.

**Supervision:** Junjie Xia, Yu Qiu, Ling Huang, Wenjun Li, Xingxiong Zou, Xiaoqian Wang, Xiaoli Hou, Yuting Chen, Haishi Xue, Runjiao Chen, Lingna Li, Jincheng Wang, Wenying Xie.

**Validation:** Junjie Xia, Yu Qiu, Ling Huang, Wenjun Li, Xingxiong Zou, Xiaoqian Wang, Yuting Chen, Runjiao Chen, Ke Yi, Jincheng Wang.

**Visualization:** Junjie Xia, Yu Qiu, Ling Huang, Wenjun Li, Xingxiong Zou, Xiaoqian Wang, Haishi Xue, Lingna Li, Ke Yi, Jincheng Wang, Wenying Xie.

**Writing – original draft:** Junjie Xia, Yu Qiu, Ling Huang, Wenjun Li, Xingxiong Zou, Xiaoqian Wang, Haishi Xue, Lingna Li, Ke Yi.

**Writing – review & editing:** Junjie Xia, Yu Qiu, Ling Huang, Wenjun Li, Ke Yi.

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
