## [Decision Letter · Decision Letter 0]

23 Sep 2024

Dear Dr. Xia,

We look forward to receiving your revised manuscript.

Kind regards,

Nishi Shahnaj Haider, Ph.D.

Guest Editor

PLOS ONE

Journal requirements: 1. When submitting your revision, we need you to address these additional requirements. Please ensure that your manuscript meets PLOS ONE's style requirements, including those for file naming. The PLOS ONE style templates can be found at https://journals.plos.org/plosone/s/file?id=wjVg/PLOSOne_formatting_sample_main_body.pdf and https://journals.plos.org/plosone/s/file?id=ba62/PLOSOne_formatting_sample_title_authors_affiliations.pdf. 2. Thank you for stating the following financial disclosure:  [Health Commission of Sichuan Province(20PJ267) and Health Commission of Mianyang(202343)].  Please state what role the funders took in the study.  If the funders had no role, please state: ""The funders had no role in study design, data collection and analysis, decision to publish, or preparation of the manuscript."" If this statement is not correct you must amend it as needed. Please include this amended Role of Funder statement in your cover letter; we will change the online submission form on your behalf. 3. We note that you have indicated that there are restrictions to data sharing for this study. For studies involving human research participant data or other sensitive data, we encourage authors to share de-identified or anonymized data. However, when data cannot be publicly shared for ethical reasons, we allow authors to make their data sets available upon request. For information on unacceptable data access restrictions, please see http://journals.plos.org/plosone/s/data-availability#loc-unacceptable-data-access-restrictions.  Before we proceed with your manuscript, please address the following prompts: a) If there are ethical or legal restrictions on sharing a de-identified data set, please explain them in detail (e.g., data contain potentially identifying or sensitive patient information, data are owned by a third-party organization, etc.) and who has imposed them (e.g., a Research Ethics Committee or Institutional Review Board, etc.). Please also provide contact information for a data access committee, ethics committee, or other institutional body to which data requests may be sent. b) If there are no restrictions, please upload the minimal anonymized data set necessary to replicate your study findings to a stable, public repository and provide us with the relevant URLs, DOIs, or accession numbers. Please see http://www.bmj.com/content/340/bmj.c181.long for guidelines on how to de-identify and prepare clinical data for publication. For a list of recommended repositories, please see https://journals.plos.org/plosone/s/recommended-repositories. You also have the option of uploading the data as Supporting Information files, but we would recommend depositing data directly to a data repository if possible. Please update your Data Availability statement in the submission form accordingly. 4. In the online submission form, you indicated that [The dataset used and analysed during the current study is available from the corresponding author on reasonable request]. All PLOS journals now require all data underlying the findings described in their manuscript to be freely available to other researchers, either 1. In a public repository, 2. Within the manuscript itself, or 3. Uploaded as supplementary information.This policy applies to all data except where public deposition would breach compliance with the protocol approved by your research ethics board. If your data cannot be made publicly available for ethical or legal reasons (e.g., public availability would compromise patient privacy), please explain your reasons on resubmission and your exemption request will be escalated for approval.  5. Your ethics statement should only appear in the Methods section of your manuscript. If your ethics statement is written in any section besides the Methods, please delete it from any other section.  6. Please include your tables as part of your main manuscript and remove the individual files. Please note that supplementary tables (should remain/ be uploaded) as separate ""supporting information"" files".

Additional Editor Comments:

Authors are requested to sincerely revise the manuscript as per the suggestions of the respected reviewers and have to submit the revised version of the manuscript along with your response to each of the reviewers comments.

Reviewers' comments:

**Comments to the Author**

1. Is the manuscript technically sound, and do the data support the conclusions?

Reviewer #1: Partly

Reviewer #2: No

Reviewer #3: Partly

2. Has the statistical analysis been performed appropriately and rigorously?

Reviewer #1: Yes

Reviewer #2: No

Reviewer #3: Yes

3. Have the authors made all data underlying the findings in their manuscript fully available?

Reviewer #1: Yes

Reviewer #2: No

Reviewer #3: No

4. Is the manuscript presented in an intelligible fashion and written in standard English?

Reviewer #1: No

Reviewer #2: Yes

Reviewer #3: Yes

Reviewer #1: First, I appreciate the authors for investigated the Prevalence, Risk Factors of Preserved Ratio Impaired Spirometry in adult in plateau. However, there are several areas of concern that need to be addressed before the paper can be indexed.

In introduction: The introduction should clearly state the need for the study.

In Page No-2, line-50: “Studies1 found that the prevalence of COPD was higher in high-altitude areas.” At least 2 or more than 2 citations should be quoted for'studies’.

In Page No-3, line-57: “Previous studies have shown that individuals with 58 PRISm…….4,5,6” –It would be good if authors included the place of study for these references.

In Page No-4, line-85: “According the guideline, we used the survey used uniform methods, procedures….”. – It would be good if the authors state the name of the guideline in the manuscript.

In Page No-6, line-157: “This study was consistent with most previous studies.” – The references should be added for most previous studies.

Reviewer #2: -The study relies on self-reported data from participants for key variables like smoking history, history of tuberculosis, and medical conditions (diabetes, hypertension). This introduces potential recall bias, where participants may inaccurately report their histories. This could especially affect data related to smoking habits and tuberculosis history, as both factors have strong social and stigma-related influences

-Although the study briefly mentions different prevalence rates among ethnic groups (Han and Tibetan), it doesn’t explore whether these differences are statistically significant or delve into the possible cultural or genetic factors influencing these rates.

Reviewer #3: The topic sounds good , and the methods sounds clear

Just some main points :

1.Sample size calculation and power analysis were not done.

2. I can not find the tables (table 1 and table 2) to evaluate the results.

**Do you want your identity to be public for this peer review?** For information about this choice, including consent withdrawal, please see our Privacy Policy

Reviewer #1: No

Reviewer #2: No

Reviewer #3: **Yes: ** Asmaa Abd Elhameed

---

## [Author Response · Author response to Decision Letter 0]

24 Oct 2024

Ethics was approved by The Third Hospital of Mianyang's Ethics Committee. As part of the ethics review process, participant confidentiality restrictions prohibit the authors from making the data set publicly available. During the consent process,

participants were explicitly guaranteed that the data would only be seen my members of the study team. For any discussions about the data set please contact The Third Hospital of Mianyang's Research Governance:578012475@qq.com

Reviewer 1:

1.In introduction: The introduction should clearly state the need for the study.

Thank you very much for recognition of our work. We really appreciated this advice.

COPD is a major public health concern, and PRISm is currently considered to be pre-COPD, which can progress to COPD in some people over time. The incidence of COPD is higher in plateau areas, especially in Asia. Therefore, understanding the prevalence and risk factors of PRISm in highland areas is helpful to prevent COPD.

2.In Page No-2, line-50: “Studies1 found that the prevalence of COPD was higher in high-altitude areas.” At least 2 or more than 2 citations should be quoted for'studies’.

We really appreciated this advice; We added two more citations for a higher incidence of COPD in high areas.

3.In Page No-3, line-57: “Previous studies have shown that individuals with 58 PRISm…….4,5,6” –It would be good if authors included the place of study for these references.

We really appreciated this advice; We added the place of study.

4.In Page No-4, line-85: “According the guideline, we used the survey used uniform methods, procedures….”. – It would be good if the authors state the name of the guideline in the manuscript.

We really appreciated this advice; We added the name of the guideline.

5.In Page No-6, line-157: “This study was consistent with most previous studies.” – The references should be added for most previous studies..

We really appreciated this advice; The references included references 9 and 16

Reviewer 2:

1.The study relies on self-reported data from participants for key variables like smoking history, history of tuberculosis, and medical conditions (diabetes, hypertension). This introduces potential recall bias, where participants may inaccurately report their histories. This could especially affect data related to smoking habits and tuberculosis history, as both factors have strong social and stigma-related influences.

We really appreciated this advice.Our questionnaire was conducted in a quiet room by a trained respiratory physician. All patient information is kept confidential and is asked based on a standardized questionnaire. We also reviewed the participants' medical records. This reduces recall bias to some extent. However, due to the backward medical conditions in Tibetan areas on the plateau, many participants' medical records are incomplete, and some patients' medical records in other places cannot be provided, which will indeed lead to recall bias. Thank you very much for the comments. In the future work, we will pay more attention to this problem and make the obtained data more authentic and credible.

2.Although the study briefly mentions different prevalence rates among ethnic groups (Han and Tibetan), it doesn’t explore whether these differences are statistically significant or delve into the possible cultural or genetic factors influencing these rates.

We really appreciated this advice.In the single factor analysis, we found that the prevalence of different ethnic groups was inconsistent, but after the multi-factor analysis, it was found that ethnicity was not an independent risk factor. That's why we didn't talk about nationalities.

Reviewer 3:

1.Sample size calculation and power analysis were not done.

We really appreciated this advice. Since there is no available research data for reference in China, we use a single proportional formula to calculate the sample size. N=Z2*P (1-P) /d2. Then the sample size is 544 based on the total population. Considering that the loss of follow-up rate was 10%, the final sample size was 560.

2. I can not find the tables (table 1 and table 2) to evaluate the results.

We really appreciated this advice.We have put Table 1 and Table 2 in the manuscript.

---

## [Decision Letter · Decision Letter 1]

11 Dec 2024

Dear Dr. Xia,

Thank you for submitting your manuscript to PLOS ONE. After careful consideration, we feel that it has merit but does not fully meet PLOS ONE’s publication criteria as it currently stands. Therefore, we invite you to submit a revised version of the manuscript that addresses the points raised during the review process.

We look forward to receiving your revised manuscript.

Kind regards,

Nishi Shahnaj Haider, Ph.D.

Guest Editor

PLOS ONE

Additional Editor Comments (if provided):

Reviewers have recommended consideration of the manuscript following major revision. Authors are expected to revise the manuscript as per the reviewers comments.

Reviewers' comments:

Reviewer's Responses to Questions

**Comments to the Author**

Reviewer #1: All comments have been addressed

Reviewer #2: All comments have been addressed

2. Is the manuscript technically sound, and do the data support the conclusions?

Reviewer #1: Partly

Reviewer #2: Yes

3. Has the statistical analysis been performed appropriately and rigorously?

Reviewer #1: Yes

Reviewer #2: Yes

4. Have the authors made all data underlying the findings in their manuscript fully available?

Reviewer #1: Yes

Reviewer #2: No

5. Is the manuscript presented in an intelligible fashion and written in standard English?

Reviewer #1: Yes

Reviewer #2: No

Reviewer #1: (No Response)

Reviewer #2: Recommendations for Improvement:

-Conduct Power Analysis:

Include a detailed power analysis to support the adequacy of the sample size.

-Address Recall Bias:

Highlight steps taken to minimize recall bias more explicitly and suggest future improvements.

-Explore Ethnic Variations:

Delve deeper into the differences observed between ethnic groups, providing statistical significance and possible explanations.

-Enhance Discussion:

Expand the discussion to include comparisons with global studies and implications for public health policy.

-Improve Data Accessibility:

Consider anonymizing the dataset to allow partial access for peer verification while adhering to ethical guidelines.

-Language and Presentation:

Proofread for grammar and typographical errors to enhance readability and professionalism.

**Do you want your identity to be public for this peer review?** For information about this choice, including consent withdrawal, please see our Privacy Policy

Reviewer #1: No

Reviewer #2: **Yes: ** Dr. Mais Odai

---

## [Author Response · Author response to Decision Letter 1]

2 Jan 2025

Answers to reviewers

Changes in the main text indicated in red

Reviewer 2:

1.-Conduct Power Analysis:Include a detailed power analysis to support the adequacy of the sample size.

Thank you very much for recognition of our work. We really appreciated this advice.

This is a cross-sectional study,since there was no available research data for reference in China, we used a single proportional formula to calculate the sample size. N=Z2*P (1-P) /d2. Then the sample size was 544 based on the total population. Considering that the loss of follow-up rate was 10%, the final sample size was 560(Z=1.96,p=0.15,d=0.2p).

2.Address Recall Bias:Highlight steps taken to minimize recall bias more explicitly and suggest future improvements.

We really appreciated this advice.Our questionnaire was conducted in a quiet room by a trained respiratory physician. The uniformly trained respiratory physicians asked the respondents face to face and asked questions in the independent and quiet room according to the questionnaire items.At the same time, we reviewed the respondents' medical records. If there was any discrepancy between the medical records and the responses, the medical records were taken as the reference.

3.Explore Ethnic Variations:Delve deeper into the differences observed between ethnic groups, providing statistical significance and possible explanations.

We really appreciated this advice.Our previous research has found that the prevalence rate of COPD in Han population was higher than that in Tibetan population, suggesting that the prevalence of COPD may be related to ethnic and racial differences. Havryk et al studied Sherpas living at an average altitude of 4000 meters, showing that their lung capacity increased by about 12% compared to Caucasians. Wood conducted lung function studies on Ladakh and Tibetans living between 3300 and 4500 meters above sea level. Spirometry results in both groups showed high values of maximal mid-expiratory ﬂow are between 130% and 150% of predicted, and a FEV1/FVC ratio of 115%. These studies suggest that, over time and over multiple generations, highaltitude populations have undergone physiological and genetic adaptations to extreme altitudes in response to chronic hypoxia and generally high-intensity exercise. Living at high altitudes is associated with accelerated lung function decline in populations with low daily smoking rates. However, the lung function of the Han population is lower than that of the Tibetan population, and is more susceptible to decline due to environmental factors.However, In this study, We found no statistically significant difference in prevalence among different ethnic groups. To find out why, prospective long-term follow-up studies may be needed.

4.Enhance Discussion:Expand the discussion to include comparisons with global studies and implications for public health policy.

We really appreciated this advice. At present, there is a lack of global data on PRISm studies in plateau areas, and the results of this study are similar to those of other studies, which found that smoking, age, BMI≥30, diabetes, history of tuberculosis, hypertension, white blood cell count, and red blood cell volume distribution width are correlated with PRISm. The purpose of this study is to supplement some of the plateau data for the global PRISm research.In highland areas, we can prevent PRISm development by controlling weight, smoking, diabetes, and high blood pressure. A history of tuberculosis may be a specific risk factor for PRISm in the plateau region of China. We can also detect PRISm early by focusing on the WBC and RDW of the population.

5.Improve Data Accessibility:Consider anonymizing the dataset to allow partial access for peer verification while adhering to ethical guidelines.

We really appreciated this advice.Ethics was approved by The Third Hospital of Mianyang's Ethics Committee. As part of the ethics review process, participant confidentiality restrictions prohibit the authors from making the data set publicly available. During the consent process,participants were explicitly guaranteed that the data would only be seen my members of the study team. For any discussions about the data set please contact The Third Hospital of Mianyang's Research Governance:578012475@qq.com.

---

## [Decision Letter · Decision Letter 2]

19 Jan 2025

Prevalence, Risk Factors of Preserved Ratio Impaired Spirometry in adult in plateau: A Cross-Sectional Study

PONE-D-24-22504R2

Dear Dr. Xia,

We’re pleased to inform you that your manuscript has been judged scientifically suitable for publication and will be formally accepted for publication once it meets all outstanding technical requirements.

Kind regards,

Nishi Shahnaj Haider, Ph.D.

Guest Editor

PLOS ONE

Additional Editor Comments (optional):

Reviewers are agreed with the revisions been submitted for all the given comments. Therefore, the decision is to 'Accept' the manuscript for publication.

Reviewers' comments:

Reviewer's Responses to Questions

**Comments to the Author**

Reviewer #2: All comments have been addressed

2. Is the manuscript technically sound, and do the data support the conclusions?

Reviewer #2: Yes

3. Has the statistical analysis been performed appropriately and rigorously?

Reviewer #2: Yes

4. Have the authors made all data underlying the findings in their manuscript fully available?

Reviewer #2: Yes

5. Is the manuscript presented in an intelligible fashion and written in standard English?

Reviewer #2: Yes

Reviewer #2: Dear Author

All recommendations have been made correctly

thank you very much

**Do you want your identity to be public for this peer review?** For information about this choice, including consent withdrawal, please see our Privacy Policy

Reviewer #2: No

---

## [Editor Report · Acceptance letter]

PONE-D-24-22504R2

PLOS ONE

Dear Dr. Xia,

I'm pleased to inform you that your manuscript has been deemed suitable for publication in PLOS ONE. Congratulations! Your manuscript is now being handed over to our production team.

Kind regards,

on behalf of

Dr. Nishi Shahnaj Haider

Guest Editor

PLOS ONE